# Community Perceptions of a Multilevel Sanitation Behavior Change Intervention in Rural Odisha, India

**DOI:** 10.3390/ijerph17124472

**Published:** 2020-06-22

**Authors:** Renee De Shay, Dawn L. Comeau, Gloria D. Sclar, Parimita Routray, Bethany A. Caruso

**Affiliations:** 1Department of Behavioral, Social, and Health Education Sciences, Rollins School of Public Health, Emory University, Atlanta, GA 30322, USA; reneedeshay@gmail.com (R.D.S.); dcomeau@emory.edu (D.L.C.); 2Gangarosa Department of Environmental Health, Rollins School of Public Health, Emory University, Atlanta, GA 30322, USA; gloria.sclar@emory.edu; 3Independent Consultant, Bhubaneswar, Odisha 751006, India; rparimita@gmail.com; 4Hubert Department of Global Health, Rollins School of Public Health, Emory University, Atlanta, GA 30322, USA

**Keywords:** WASH, qualitative research, latrine use, process evaluation, gender, collective efficacy

## Abstract

While latrine coverage is increasing in India, not all household members use their latrines. Cost-effective, culturally appropriate, and theory-informed behavior change interventions are necessary to encourage sustained latrine use by all household members. We qualitatively examined community perceptions of sanitation interventions broadly, along with specific impressions and spillover of community-level activities of the Sundara Grama latrine use behavior change intervention in rural Odisha, India. We conducted sixteen sex-segregated focus group discussions (*n* = 152) in three intervention and three nonintervention villages and thematically analyzed the data. We found Sundara Grama was well-received by community members and considered educative, but perceptions of impact on latrine use were mixed and varied by activity. Intervention recruitment challenges prevented some, such as women and households belonging to lower castes, from attending activities. Spillover occurred in one of two nonintervention villages, potentially due to positive relations within and between the nonintervention village and nearby intervention village. Community-level sanitation initiatives can be hindered by community divisions, prioritization of household sanitation over community cleanliness, and perceptions of latrine use as a household and individual issue, rather than common good. Community-centered sanitation interventions should assess underlying social divisions, norms, and perceptions of collective efficacy to adapt intervention delivery and activities.

## 1. Introduction

Open defecation is associated with diarrhea, parasitic infections, malnutrition, and child stunting [1,2,3,4,5]. Poor sanitation conditions can affect an individual’s mental well-being and personal safety, especially for women and children, who risk bodily exposure, harassment, and violence when defecating in the open [6,7,8,9]. In contrast, improved sanitation is protective against diarrhea [4,5], soil-transmitted helminth infections [4], active trachoma [4], intestinal protozoa infections [2] and schistosomiasis [3], and is also associated with improved cognitive development [10] through a reduction in exposure to fecal pathogens that cause diarrhea and environmental enteropathy [11,12,13,14].

Despite benefits of sanitation, sustained use of latrines remains a challenge, particularly in India [15]. The 2019 WHO/UNICEF Joint Monitoring Programme (JMP) for Water Supply, Sanitation and Hygiene report estimated that 26% of India’s population practiced open defecation, 2% used an unimproved facility, 13% used a limited (shared) sanitation facility, and 60% used at least a basic sanitation facility. Open defecation is most frequently practiced in rural areas (36%) [16], and as per India’s National Family Health Survey (2015–2016), collected just prior to the initiation of the present study, only 37% of rural households reported using an improved sanitation facility [17]. Comparatively, data from the WHO/UNICEF JMP report and World Bank showed that neighboring countries Bangladesh and Pakistan, despite having lower GDPs [18], had lower open defecation rates (<1% and 10%, respectively) [16]. Pakistan and Bangladesh had equal or lower percentages of their population using at least basic sanitation services compared to India (60% and 48% respectively), but had higher percentages of their populations using either an unimproved or limited sanitation facility [16].

Studies have found that the numerous efforts by the Government of India to increase latrine coverage have resulted in only modest reductions in open defecation, and no impact on child health [19,20]. Since the mid-1980s, the Indian government has implemented country-wide sanitation programs, gradually shifting from solely constructing latrines to also promoting latrine uptake. The Total Sanitation Campaign (1999–2012) introduced information, education, and communication (IEC) campaigns and provided latrine construction subsidies [21], but trials found no improvement in health outcomes, citing suboptimal uptake as a potential explanation [19,20,22]. These findings inspired a call for increased investment in sanitation behavior change efforts [23].

Initiated in 2014, the latest sanitation campaign, the Swachh Bharat Mission (SBM), recognized the need for IEC activities, but limited IEC spending to only 8% of the budget and did not dedicate specified personnel to execute IEC activities [24]. An audit of fiscal year 2016–2017 found only 1% of the budget had been spent on IEC, increasing to only 4% by fiscal year 2018–2019 [25]. The key difference between SBM and previous campaigns is the scale of the campaign’s goal—to make India open defecation free by October 2, 2019 [26,27], which was indeed declared by Prime Minister Narendra Modi on said date [28]. Experts and the media have questioned the accuracy of this claim [29,30,31,32].

Interventions aimed at increasing latrine coverage in India have not necessarily led to increased use [15,33]. Sociocontextual and behavioral factors associated with latrine use and non-use need to be understood and addressed to increase and sustain uptake [34,35,36]. Barriers to latrine use include improper or incomplete latrine construction, lack of a nearby water source for postdefecation cleansing, cultural beliefs around purity and pollution, fear of rapid pit filling, and strong preference for open defecation [34,35,36,37,38].

Researchers at Emory University designed Sundara Grama, a multilevel behavior change intervention, to increase latrine use and safe disposal of child feces among rural, latrine-owning households in Odisha state, which had the second-lowest latrine coverage level (29%) in India in 2016 [39]. The intervention addressed six barriers to latrine use identified via formative research: (1) nonfunctional latrines; (2) limited practical knowledge of how to use a latrine and empty its pit; (3) preference for open defecation due to attitudinal and sociocultural factors; (4) limited recognition of the health and nonhealth benefits to latrine use; (5) lack of hardware to aid safe disposal of child feces; and (6) limited knowledge regarding child feces disposal [40]. Intervention villages received a series of intervention activities designed to address these barriers delivered at the community, group, and household levels. Appendix A provides descriptions of all activities.

In 2018, with SBM still underway, the Sundara Grama intervention was delivered by the local NGO Rural Welfare Institute (RWI), and evaluated by Emory. The cluster-randomized trial involved 66 villages, including 33 that received the intervention and 33 that served as controls [41]. After accounting for an increase in the control group’s latrine use, the endline evaluation found a 6.4% (95% confidence interval (CI) 2.0–10.7%, *p* = 0.004) increase in reported latrine use for persons age five and above in the intervention group [40]. Beyond measuring impact, it is critical to understand what community members thought of the intervention activities, their perceptions of sanitation programs more broadly, and if community members in nonintervention villages adjacent to intervention villages benefitted from or were influenced by the intervention. Such research helps explain Sundara Grama’s observed impact on latrine use, and offers insights for adapting and scaling the intervention in the future.

Thus, this research aimed to: (1) understand community member perceptions of sanitation interventions broadly, (2) assess community member perceptions of three of the community-level Sundara Grama intervention activities, specifically the palla performance, transect walk, and community meeting, and (3) determine possible spillover of the intervention into nonintervention villages.

## 2. Materials and Methods

### 2.1. Study Design and Setting

This qualitative study took place within the context of the larger Sundara Grama process evaluation, which involved 72 rural villages (66 in the trial, six for qualitative research) across three administrative blocks in Puri district of Odisha state [41]. The Sundara Grama intervention was delivered to 36 villages (33 trial, three qualitative research). The six qualitative villages were purposively selected, with one pair from each block and covering a range of village sizes (small, large, average) to capture perspectives from these different contexts. Two of the village pairs were also purposively selected due to their proximity to each other, enabling exploration of intervention spillover. Focus group discussions (FGDs), which enable participants to interact and prompt conversations from the emic perspective [40], were used to elicit community members’ perceptions of three community-level intervention activities: palla, transect walk, and community meeting. Sixteen sex-segregated FGDs (eight with men, eight with women) were conducted across the six selected villages (5–15 participants each; 72 women and 80 men). FGDs were sex-segregated to facilitate open discussion among group members. Data were collected in July 2018, a few weeks after the community-level activities took place.

### 2.2. Study Participants

Village members were eligible to participate if they were 18 years or older and provided consent. In intervention villages, members also needed to have attended more than one of any of the intervention activities to be eligible.

Recruitment was conducted in two ways. Primarily, the Anganwadi worker (female worker who runs the Anganwadi, a government run village preschool) or the ward member (local elected representative) identified and recruited participants in advance of the FGDs. In some cases, participants were recruited on the day of the FGD through convenience sampling by the research assistants (RAs), who went door-to-door asking for volunteers, or approached groups gathered in public spaces. Most participants were General caste and the majority owned functional latrines. Table 1 shows the demographic breakdown of participants.

### 2.3. Data Collection

FGD guide development was informed by the Sundara Grama Intervention Manual [42], and the process evaluation elements satisfaction, reach, recruitment, and context, as described by Saunders et al. (2005) [43]. Appendix A provides a brief definition of each element with example questions from the FGD guides. Two guides were developed: one for intervention villages and one for nonintervention villages. Both FGD guides included questions about what participants thought about their village, what they knew about the intervention activities, and their perceptions of the intervention’s impact on latrine use and other sanitation practices in their village. Although both guides included detailed questions about the recruitment, reach, and content of each activity, RAs were instructed to skip this section in nonintervention villages if participants had no knowledge of the intervention. The intervention village guide was piloted in a village where the RWI implementing team piloted Sundara Grama activities during their training. After five FGDs, the guides were revised to more deeply examine topics based on findings generated to that point. Specifically, questions were added to the nonintervention village guide to explore the history of sanitation programs in the village and gain understanding of the local context and sanitation practices.

FGDs were facilitated by five RAs fluent in Odia, the local language. FGDs with women were mostly held at the Anganwadi center, while FGDs with men were either held at the village temple or clubhouse, depending on the participants’ preferences. Before each FGD, a brief paper survey collecting demographic information was completed by each participant. FGDs were audio recorded, and RD (primary author) and a team member also took detailed observational notes and debriefed with the RAs daily to identify discussion themes. Recordings were subsequently uploaded to shared folders on Box, a secure storage platform used by Emory University.

### 2.4. Data Management and Analysis

Recordings were directly transcribed to English by hired translators. The transcripts were coded and analyzed in MaxQDA software v. 18.2.3 (VERBI Software. Consult. Sozialforschung. GmbH, Berlin, Germany) [44]. Demographic survey data were entered into Microsoft Excel, randomly spot-checked, and descriptively analyzed.

FGD data were analyzed using thematic analysis [45]. Transcripts were reviewed and memos written to record first impressions of the data. Data were then coded and analyzed using a “bottom-up” inductive approach [45]. A codebook was developed based on deductive codes from questions in the FGD guides, as well as inductive codes identified by reviewing the transcripts. BAC, GDS, and other colleagues gave feedback on the codebook. After the codebook was revised, the transcripts were re-coded, and codes were then grouped into themes. Themes that were aligned with the research questions were selected and analyzed further. Themes were reviewed by village and by participant sex to gain a sense of common patterns among these subgroups.

This study was approved by the Institutional Review Board at Emory University in Atlanta Georgia (00098293) and the Ethics Review Committee at Xavier Institute of Management in Bhubaneswar, Odisha. Prior to data collection, all participants provided verbal consent, and each was given an information sheet describing the study objectives and the benefits and risks to participating. Identifying data, such as names, were not collected. Participants were not compensated for participation.

## 3. Results

We present participants’ perceptions of (1) sanitation and sanitation programming, referred to as the community context for sanitation, (2) the three community-level Sundara Grama activities: palla, transect walk, and community meeting (from participants in intervention communities), and (3) intervention spillover of into nonintervention villages (from participants in communities that did not receive the intervention).

### 3.1. Community Context for Sanitation

Sanitation was considered an individual and household-level priority, but not a common good. Participants expressed doubt that their village would be able to improve sanitation and latrine use due to conflicting interests across village divisions and a perceived lack of resources and authority to do so. As a result, participants expected outsiders, whom they perceived to have resources and authority, to provide sanitation programming and encourage use.

#### 3.1.1. Prioritization of Household Sanitation

Individual and household sanitation were important for participants in all villages, so much so that village sanitation was described as the sum of each household’s and individual’s sanitation, rather than conceptualized as a broader community idea: “If you yourself are not clean, then the village also cannot be clean. But if you yourself are clean, then the village can also be cleaned and everything else can also be cleaned,” (man, nonintervention village).

Prioritizing household sanitation, however, often resulted in practices that kept the household clean, but compromised the cleanliness of the village environment. Villagers practiced open defecation in fields and threw waste, including bags, plastic bottles, and materials used to clean child feces, outside the household compound into the open peripheral spaces. One woman’s description of cleaning her child after defecating illustrates the contrast between the treatment of the household and the environment beyond: “We wash her with Dettol [commercially available liquid cleaning agent] and the floor is cleaned with phenol [household cleaner]. Then, we clean her with a cloth and throw that cloth into the lake,” (nonintervention village).

#### 3.1.2. Village Divisions

Sociocultural and geographic divisions existed within the study villages, as evidenced by material wealth disparities between hamlets (village sections), where members of different castes lived. Participants in four villages used the terms “this side” and “that side” to describe the village areas, suggesting their association with specific locations in the village, which may be further linked to caste identity. Participants also used this language to reference the sanitation behaviors of others: “People of that side are making the place filthy, otherwise this side is okay,” (man, nonintervention village). They noted how these divisions obstructed change and development in their villages, describing their villages as unable to reach consensus, resistant to improvement initiatives, and burdened by corrupt leadership and favoritism that hindered progress. In one village, attempts to clean a pond were met with violence and demands for money from the panchayat (lowest level of governing unit). In two other villages, participants complained that the Anganwadi worker withheld information about community events, leaving some to feel purposely excluded.

#### 3.1.3. Perceived Community Ability to Address Sanitation

Participants listed a variety of ways they wished to improve village cleanliness, including cleaning ponds and roads, removing trash, and addressing drainage issues, but encouraging latrine use and ending open defecation were rarely mentioned. Furthermore, some questioned the relevance of latrine use to a discussion of village sanitation when brought up by a participant sharing how the intervention influenced him to stop open defecating.

Participants reported feeling limited in their ability to influence others in their village to stop open defecating. They explained that fellow villagers were sensitive to being told how and where to defecate, and might retaliate with harsh words or physical violence. Participants expressed doubt in their authority to ask fellow villagers to practice latrine use, especially when they lacked the resources to provide nonowners with latrines: “Those who don’t have a latrine, they will counter us and ask us for a latrine. How can we stop them from open defecation, when we can’t provide them a latrine?” (woman, nonintervention village).

Women experienced additional challenges in promoting latrine use due to gender norms. One woman explained: “If a man tells them, they might listen, but if a woman tells them, they won’t give a damn,” (nonintervention village). Another woman said they may only be able to influence their family members: “We can tell our family members, but cannot tell other households […] Respective family members can tell their families,” (intervention village).

Village representatives also experienced resistance when advising against open defecation, including a ward member who tried to persuade people to stop open defecating: “As I did it, these open defecators started scolding me…Since that day, I have decided not to say anything,”(man, nonintervention village).

#### 3.1.4. Outsider Involvement in Village Sanitation

Participants across the study villages expected outsiders, such as the government, NGOs, and contractors, to support village sanitation needs, including latrine construction, sanitation awareness campaigns, and latrine use enforcement: “We are waiting for when the Government will give us the money to build a [latrine], and we haven’t got it yet. Hence, we can’t build a [latrine],” (woman, nonintervention village). Although participants had expressed doubts about their own ability to promote latrine use, they viewed outsiders as influential and capable of promoting latrine use: “If outsiders can go to our backyards and clean our household surroundings, then the household members will learn something good. If government people can come and do some sensitization programs, then things might improve,” (man, nonintervention village). Participants also described how outsiders bring increased legal authority to “create fear” through the threat of fines, penalties or revoked ration cards. One woman noted: “People should be pressurized that the district collector would come, then people will develop some fear and try to change,” (intervention village). The district collector is the administrative head of the district and has the authority to impose fines or penalties against organizations and individuals for not adhering to the rules and regulations, such as the regulations in SBM.

While participants expressed dependence on outsiders, they also identified drawbacks. Outside entities also were viewed as not dependable and frequently dishonest, providing substandard latrines, and offering things which people sometimes found not useful, like awareness and behavior change programs. Participants perceived these awareness activities as repetitive and ineffective without government subsidies for sanitation and water construction.

### 3.2. Perceptions of Sundara Grama Impact and Community Activities

Perceptions of the intervention’s overall impact on latrine use varied by village and between women and men. In one intervention village, women said that some had stopped open defecating while others had not. Women from this village also described the effect of Sundara Grama in terms of household cleanliness: “Yes, each household is cleaner now.” In contrast, men from the same village said the primary impact was increased awareness. In another village, men believed the intervention had resulted in a decrease in open defecation, to which women agreed, though they were less willing to comment on the actions of others. In the third village, women perceived a decrease in open defecation but were skeptical if the change would sustain while men felt the intervention was unnecessary because they were already latrine users. The men said the intervention should have focused on a nearby lower-caste hamlet instead of their village.

#### 3.2.1. Palla

Satisfaction with the palla was high; participants enjoyed the palla because it was entertaining, educational, and contained old, familiar stories. One woman noted, “We found the palla very funny. We enjoyed it,” while a man stated “[In the palla], they had talked about keeping the village clean. I liked hearing about that.” Of the six skits presented in the palla, participants cited two that they especially appreciated: a skit about Sriya, a scavenger whose cleanliness attracted the goddess Laxmi to visit her house with blessings, and a skit about Uncle Nidhi and the three assassins, Mouse, Mosquito, and Fly, explaining how the fly is the deadliest assassin because it spreads disease from feces. Men in two of the villages identified these as old stories they had read as children. Some men felt the repetition of the themes was useful: “Nothing was new; one good thing was they reminded us through their stories.” Some participants wanted more palla performances, but a few disagreed: “No [more] activities are needed; we have now learnt from the palla” (man).

Opinions of the palla’s impact on latrine use were mixed. In two of the men’s FGDs, participants felt the palla had a positive impact on latrine use or at least had increased sanitation awareness. One woman felt the intervention had influenced some, but not all: “Some might have liked it, some may not. But, what is the point in doing the palla, when people continue with the same practice?” Men from the third village stated that most of their villagers were already latrine users and the palla could have impacted only those belonging to lower castes: “No, no, this did not have an impact on all [sections of the village]. Only the harijans (lower/scheduled caste) were impacted by it,” (man).

Factors hindering the reach, or attendance, of the palla included poor recruitment, venue location, time of day, and weather. Recruitment efforts for the palla were not effective in all hamlets of the village. For example, women from a lower caste hamlet in one village complained they missed the palla because the Anganwadi worker had not informed them about it. Despite this, almost all FGD participants from intervention villages knew about the palla and understood its purpose to raise awareness about sanitation. Many participants that could not attend the palla learned about it from relatives or heard noises from the performance, which carried throughout the village. The venues for the palla were not always ideal for attracting and accommodating large crowds and the timing was challenging for some to attend. Participants in one intervention village felt the heavy rains negatively impacted both palla attendance and performance quality.

#### 3.2.2. Transect Walk

The early morning transect walk through the village and open defecation fields produced a varied and emotional response. Participants who favored this activity cited its novel approach of placing colored powder on feces to raise awareness of the “temples of shits.” Women in one village perceived, though inaccurately, that the outsiders (RWI implementers) had done them a service by sanitizing the feces with “medicine.” More than the other two community-level activities, participants’ descriptions of the transect walk emphasized the presence of outsiders as both good and a source of shame. One man described, “[The transect walk] was the best. With that, we could know that some outsiders have come.” In contrast, a woman stated, “How can we feel good? Outsiders came and showed us to sprinkle holi powder. Is that not shameful?” All unfavorable reactions centered on the shame it brought as outsiders exposed the dirtiness of their villages or when someone was caught in the act of open defecation. One man noted, “As the group walked towards the field, I was scared that we would see some open defecators doing the act. That would be shameful.” In that village, participants reported that a woman was caught defecating in the field and was threatened with losing her ration card, though it was unclear who made the threat. In one FGD with men, participants were surprised and shamed by the walk because red colored powder was used to mark the feces, a color also used for religious practices. This was a mistake by the implementing team; activity instructions stated that red and orange powder should not be used.

Perceived impact of the transect walk on latrine use was mixed: some participants reported that it immediately led to less open defecation, while others said any impact was short-lived or dependent on the individual. One woman reported that people had stopped defecating near her house, whereas other participants said open defecation stopped for a short time only and resumed after a few days. One woman felt attendees’ behaviors were based on their reaction to the walk: “Those who liked the walk have changed their habit and are using the latrine. But the ones who criticized came up with excuses for not using the latrine, like, ‘there is no latrine, there is no water, all these.’”

The transect walk appeared to be effective in reaching and gaining the attention of a large proportion of community members. Knowledge of the activity even reached some who did not attend the walk. Participants reported being informed about the transect walk through an announcement at the end of the palla and by the bells rung by RWI implementers the morning of the walk, signaling the beginning of the activity. Some women missed the announcement because they were not at the palla, but some were still able to participate because they heard the bells. Many initially interpreted the bell sounds to signal a religious event. One woman feared they had done something wrong: “We got scared when they told us to be present early morning. We felt that we probably had made some mistake, for which we were being called in the early morning.”

#### 3.2.3. Community Meeting

Of the three activities, participants had the least to say about the community meeting and mostly expressed ambivalence. Men in two villages said the community meeting was good, though it contained no new information for them and they wished the meeting had brought them latrines. One man shared: “It was good. If households are given more latrines, especially the bigger families, then people would have benefitted more.” Women in one village shared their latrine problems in the community meeting and, like the men, were disappointed the intervention did not include latrine construction.

It was unclear if the community meeting impacted latrine use. However, some women appreciated the community meeting as a vehicle for change. One woman noted: “We liked the idea that whatever we don’t like, we can make changes. If something is not clean, then we can clean it.” Aside from latrine-related change, the women in this FGD said the village development club had done other activities as a result of the intervention, including planting trees, removing weeds, cleaning the village, and maintaining peace.

The community meeting activity may not have effectively reached as many women as men. In one village, participants discussed the low attendance of women and described how some women had to leave early: “A few women were told to leave, as they had to prepare their lunch,” (man).

### 3.3. Intervention Spillover between Villages

While participants in both of the paired nonintervention villages were generally aware of the intervention, participants in nonintervention village A were more acutely aware of the intervention’s key messages and activities. Both men and women in this village understood that Sundara Grama was a sanitation intervention, but while men were only aware of the palla, the women knew about all three community activities. In contrast, participants from nonintervention village B could only describe hearing the drums or the sound system from the palla.

The quality of the relationships within and between the intervention and its paired nonintervention village appeared to influence intervention spillover into the nonintervention village. Participants from nonintervention village A and its paired intervention village described relationships within the villages as good, and that information about the intervention spread to village A through various social networks, including acquaintances, family members, children at school, and passersby. In contrast, participants in nonintervention village B and its paired intervention village described conflict and tensions both within and between the villages. Men in nonintervention village B specifically blamed their lack of knowledge about the intervention on their poor relationship with the intervention village: “No, we aren’t on good terms with them, so we don’t know what happened or not, in that village.”

Spillover was specifically reported for the transect walk activity, by women in nonintervention village A. The transect walk produced strong emotional and behavioral responses from the women, who expressed feeling shame for their village as a result of the activity occurring nearby. One woman explained, “Our village’s name got exposed, and outsiders got to know that people are defecating on the roadside.” Some women claimed implementers had scolded open defecators and that someone had video recorded the transect walk and posted it on the internet. Like in the intervention villages, women’s perceptions of the intervention’s impact were mixed. Some felt the transect walk had impacted behavior: “Yes, people have changed, like those who had stored firewood [inside their latrines]—they have removed them after this incident and are using the latrine.” Others argued behavior had not changed: “But still people remain the same, and have not changed. Recently, I saw a man was defecating on the road side in the broad daylight.”

## 4. Discussion

This research aimed to understand community perceptions of sanitation interventions in general and Sundara Grama’s three community-level intervention activities specifically, as well as intervention spillover. We found sanitation is not necessarily viewed as a common good; instead, household-level cleanliness is prioritized. Participants lacked confidence in their ability to influence the sanitation practices of fellow village members. Sundara Grama’s community intervention activities were generally viewed positively and were thought to have increased awareness around the importance of latrine use and renewed interest in village cleanliness. However, participants expressed mixed feelings around the effectiveness of the intervention to impact latrine uptake and sustained use and also reported issues around activity reach. These qualitative findings align closely with the larger trial, which found a significant, but limited, increase in latrine use. Spillover occurred to varying degrees and appeared dependent upon intra- and intervillage dynamics.

Despite India being declared open defecation free in October 2019, the study findings hold importance for the post-SBM era. Sanitation behavioral interventions are still needed to maintain latrine adoption and address gaps in coverage and use, which are documented and argued to still exist throughout India [15,29,32]. Moving forward, there remains a need to focus on maintaining latrine use behavioral gains to prevent ‘slippage,’ or reversion to open defecation. A systematic review of factors that sustain adoption of water, sanitation and hygiene (WASH) technologies found that frequent personal contact from health promoters was the most important factor influencing sustainability [46]. Our findings also found participants to be interested in repeated behavior change messaging; many study participants appreciated the Sundara Grama messages because, even though they were not new, they acted as important reminders. While the trial showed limited increase in latrine use, it is possible that Sundara Grama helped sustain latrine use by reminding villagers of what motivates them to use the latrine, or what motivated those that SBM alone was not influencing. Some participants found the intervention could not change sanitation behavior if improvements were not also made to sanitation facilities and water access. Indeed, behavior change interventions should take into account the physical needs of the community and determine if and how these needs can be met to enable behavior change. Latrine repairs were a part of the Sundara Grama intervention package because not all latrine owners had latrines that were functional or private to enable use. These had not been completed at the time of this study to determine if repairs made were deemed suitable.

Opportunities to improve Sundara Grama delivery were identified, and these lessons can be applied to other sanitation interventions moving forward. Specifically, we found that households belonging to lower castes and women sometimes faced barriers to accessing Sundara Grama activities. A previous study of community mobilization for latrine promotion in Odisha identified similar logistical challenges to gathering community members of different castes and sections and engaging women in activities [47]. Though barriers may differ in other contexts, implementers of community interventions—in India and beyond—should ensure that recruitment extends to all; multiple and varied recruitment efforts are recommended. Further, activity venues and times should be accessible to all, or activities should be held in multiple venues and at different times to accommodate diverse groups. Separate activities for men and women also may improve reach of and engagement with intervention activities. Studies in India and Kenya have shown that women are less likely to speak up during community meetings if men are present [47,48]. In Sundara Grama, while the community meetings were intended to be sex-segregated, budget constraints allowed for only one co-ed meeting.

Our qualitative work identified the transect walk as having the potential for shaming. Women in rural India typically defecate very early in the morning or after dark [34,49,50], which may put them at risk of being shamed or harassed if seen during the walk [51]. Changing the time of the transect walk from early morning to later in the day would help protect women while still enabling identification of feces in the environment. Shifting the time also could reduce the potential for punitive action against open defecators, particularly for individuals belonging to lower castes who are more likely to experience coercion and violence in response to pressure to reach SBM goals [15,31,52].

Recent WASH trials have shown little impact on health [53,54,55], leading to calls for “transformative WASH” approaches that “radically reduce fecal contamination in the household environment of low and middle-income countries” [56] (p. e1145). Community-level interventions that call upon the broader community to take action may be needed to truly eliminate, or at least reduce, fecal contamination in the environment. New research has demonstrated the need to assess two critical assumptions before implementing a community-level public health intervention based on collective action: (1) target participants believe their community is capable of taking collective action and (2) participants agree that the proposed collective action is an appropriate response to their common interests [57]. Our findings show that these assumptions may not have been met before the Sundara Grama intervention was implemented and, as a result, may have contributed to the limited impacts the trial found on latrine use. We found that while our study communities aspired to improve their village’s state of sanitation through actions like pond clean up, participants expressed low confidence in their ability to specifically curtail open defecation, despite activities like the community meeting, which encouraged collective action. Participants believed that they lacked the ability and the authority to compel or influence others (especially non-latrine-owners) to use a latrine. Along with low perceptions of collective efficacy, we found community members did not conceptualize latrine use as a behavior that contributes to their village’s sanitation and thus requires collective action. Latrine use was instead viewed as a more personal behavior that pertains to household and personal cleanliness—a finding documented by other studies in India [37,38,58].

Discourse around collective efficacy is growing in the WASH sector and new research provides insights into how researchers and practitioners can better understand the collective characteristics of the communities they engage, and subsequently design more effective community-based programming globally. Delea, et al. (2018) have devised a scale to measure perceptions of collective efficacy, which includes factors like social disorder, social response, common values, social capital, social equity, community attachment, and agency, as potentially influential to the uptake of WASH behaviors [59,60]. Such a measure could help WASH program planners to better design, target, and evaluate their community-based interventions, although the measure may require adaptation to the local culture [57].

Along with collective efficacy, specific focus might be needed on social cohesion within the WASH sector. Community-Led Total Sanitation (CLTS) is a well-established community-based sanitation intervention that relies on communities to collectively act in order to end open defecation. The CLTS Handbook states that the approach is more likely to succeed if implemented in communities that may already be socially cohesive (as indicated by small community size, homogenous population, a strong tradition of joint action, etc.) [61]; a statement supported by a multicountry follow-up evaluation of different CLTS interventions that identified social cohesion as a factor in achieving sustained sanitation outcomes [62]. Correspondingly, previous research has shown that divisions and conflict caused by social hierarchies, caste, and religious differences are prevalent in Indian villages and can negatively impact sanitation efforts [63,64]. Our study found evidence of social conflict and division among participants, which influenced the perception of their ability to improve their village and inhibited information flow to other villages. Despite these divisions, villages that received the Sundara Grama intervention succeeded in carrying out some improvement activities, such as planting trees and removing weeds, and expressed a desire to do other activities, such as cleaning ponds and fixing roads. It seems possible that if future iterations of Sundara Grama included a series of community meeting-type activities, it may be more successful in fostering the view that latrine use deserves collective action. Behavior change sanitation interventions will need to assess social cohesion in target communities to determine whether it is feasible to mobilize the community to tackle the problem, and if not, adopt a group or household-focused approach.

Building social cohesion and social capital may also increase intervention spillover within the village, for those who could not attend activities, and to nearby villages. The women’s FGD in nonintervention village A was able to provide much more information about the intervention than the men’s FGD in the same village. This could be because women are less likely to spend their days working outside their village and therefore may be more plugged in to the village social network. One study in rural India has suggested that social networks may be a more effective target for public health interventions than village units [65]. To leverage social networks, interventions could identify and target established groups that have broad reach within the community as a means of diffusing messages to those who may be hard to reach. Such a tactic may be especially useful for reaching women who identified barriers to direct participation in intervention activities. Though community-level interventions may still be needed to achieve transformative WASH outcomes, more research into optimizing diffusion of interventions through social networks may help to fill in the gaps in intervention coverage and reach.

### Strengths and Limitations

A key strength of this work was that we designated and engaged six villages—in addition to the 66 in the larger trial—to enable assessment of perceptions of the intervention shortly after intervention delivery. This engagement allowed us to learn about participants’ experiences and perceptions of the intervention activities and inform our understanding of trial findings, without disrupting the trial itself. One intervention and one nonintervention village was selected from each of the three targeted blocks, covering the full geographic context of where Sundara Grama was implemented. Although data was collected from only three intervention villages, responses were fairly consistent and aligned with the trial findings, suggesting they are reliable.

Recruitment for the study was done primarily through the ward member and Anganwadi worker, who may have been biased in their selection of participants. However, our secondary recruitment method of the convenience sampling by RAs gave us access to other participants and may have increased the diversity of our sample. Participants tended to be older (average age of all participants = 45.8 years, SD = 13.3), with fewer participants from other age groups. This may have been influenced by the timing of the FGD, when younger adults might be busy working. The majority of participants (65%) were from General caste, which is likely not representative of village demographics, based on the larger trial data which found only 36% of households were General caste [40]. Household latrine ownership among participants (72.4% owners) was slightly higher than the larger trial data, which found an average of 63.7% of households in intervention villages and 67.3% of households in control villages owned latrines [40]. This may be because of the higher percentage of General caste members in our sample.

Finally, as this study took place during SBM, it was not always clear if participants differentiated Sundara Grama activities from other sanitation events that could have taken place in their village. Participants occasionally brought up events that were not part of Sundara Grama, such as an SBM rally, and outsiders sharing photos and videos of open defecators. This illustrates the importance of understanding the cultural context and history of sanitation programming in target communities and potential for shame and coercion when designing and implementing sanitation interventions, especially in post-SBM India.

## 5. Conclusions

There is still much work to be done in the post-SBM era to maintain previous behavioral gains and address remaining gaps in latrine coverage and use. Our findings show that it is critical to understand communities’ perceptions of past sanitation interventions and level of collective efficacy in order to improve effectiveness of future programs. Sundara Grama participants spoke positively about the intervention, but caste divisions, social conflict, and gender norms kept many villagers from fully experiencing the activities as intended. Future sanitation interventions in India should take into consideration barriers to reach, engage robust recruitment strategies to enable participation of women and members of all castes, and ensure activities do not result in shame or harm, even if unintended.

## Figures and Tables

**Table 1 ijerph-17-04472-t001:** Demographic Information of Focus Group Participants in Total, and by Sex.

Participant Characteristics	Total		Female		Male	
	N (%)		N (%)		N (%)	
Total participants	152		72	(47.4%)	80	(52.6%)
Intervention villages	65	(42.8%)	35	(53.8%)	30	(46.2%)
Nonintervention villages	87	(57.2%)	37	(42.5%)	50	(57.5%)
Average age in years	45.8	SD = 13.3	43.3	SD = 11.0	48.0	SD = 14.9
Caste ^1^	152					
General Caste	99	(65.1%)	45	(45.5%)	54	(54.5%)
Other Backward Castes	37	(24.3%)	18	(48.6%)	19	(51.4%)
Scheduled Caste	13	(8.6%)	6	(46.2%)	7	(53.8%)
Scheduled Tribe	3	(2.0%)	3	(100%)	0	(0%)
Latrine ownership ^2^	152					
Yes, functional	101	(66.4%)	43	(42.6%)	58	(57.4%)
Yes, nonfunctional	8	(6.0%)	5	(62.5%)	3	(37.5%)
No	42	(27.6%)	24	(57.1%)	18	(42.9%)
Frequency of latrine use for defecation (among latrine owners)	110					
Always	96	(87.2%)	42	(43.8%)	54	(56.3%)
Sometimes	10	(9.0%)	3	(30.0%)	7	(70.0%)
Never	4	(3.6%)	3	(75.0%)	1	(25.0%)
Attended the *palla* (intervention villages only)	59					
Yes	32	(54.2%)	16	(50.0%)	16	(50.0%)
No	27	(45.8%)	19	(70.4%)	8	(29.6%)
Attended the transect walk (intervention villages only)	58					
Yes	30	(51.7%)	23	(76.7%)	7	(23.3%)
No	28	(48.3%)	12	(42.9%)	16	(57.1%)
Attended the community meeting (intervention villages only)	55					
Yes	31	(56.4%)	21	(67.7%)	10	(32.3%)
No	24	(43.6%)	13	(54.2%)	11	(45.8%)
Attended the mother’s group meeting (intervention villages only)	43					
Yes	23	(53.5%)	21	(91.3%)	2	(8.70%)
No	20	(43.6%)	11	(55.0%)	9	(45.0%)
Attended the household visit (intervention villages and latrine owners only)	23					
Yes	20	(87.0%)	6	(30.0%)	14	(70.0%)
No	3	(13.0%)	3	(100%)	0	(0%)

^1^. For consistency, we have used government terminology to describe the participants’ castes. ^2^. One missing value among latrine owners; unknown if one latrine was functional or nonfunctional.

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
