# Peer review of "Community Perceptions of a Multilevel Sanitation Behavior Change Intervention in Rural Odisha, India"

_ijerph, 2020, doi:10.3390/ijerph17124472_

Round 1
Reviewer 1 Report
I have several comments for the authors
1) This is a very important and interesting research from a public health standpoint. While the findings were scientifically relevant and insightful, to what extent they could be extended to economically deprived regions of the world. I think the authors should endeavor to incorporate that in their manuscript if that is the case.
2) Lines 150-152: I was wondering if the collected and analyzed data were normally distributed. Could this be indicated in the article?
3) On the average, how many interventions that participants have attended and in the same vein the contrast between man vs woman. Would this have an impact on the overall acceptance and perception of latrine use.
It was an interesting research article and I greatly enjoyed reading it.
Regards,
Author Response
15 June 2020
To Whom It May Concern:
RE: ‘Community perceptions of a multi-level sanitation behavior change intervention in rural Odisha, India’(manuscript: ijerph-831499).
Many thanks to the reviewers who took the time to review our manuscript and give their feedback. Below is a point-by-point response to all comments.We have copied refered text sections into the responses below with line numbers, underlined all edits within text sections, and have also made all changes in track changes on the manuscript document for reviewing ease.Upon rereview, we have also made a few small edits to the manuscript.
We hope these revisions are met with your approval and appreciate your consideration of this manuscript. Please contact me if you have any questions.
Sincerely,
Bethany A. Caruso
REVIEWER 1
Comments and Suggestions for Authors
1) This is a very important and interesting research from a public health standpoint. While the
findings were scientifically relevant and insightful, to what extent they could be extended to
economically deprived regions of the world. I think the authors should endeavor to incorporate
that in their manuscript if that is the case.
Response from authors:
First, we thank reviewer 1 for the time provided to review this manuscript, as well as the overall positive reception.
Regarding the suggestion to provide insights as to how the learning could be extended to economicaly deprived regions of the world:
Many of our discussion points do speak to specific socio-cultural barriers within rural Indian villages, namely the caste system, specific gender norms, and cultural beliefs about purity, and how they affect community-level collective action interventions because this study was conceived in India in the context of SBM and the unique challenges India has faced in its journey to become open defecation free.
Still, we deliberately have made efforts to indicate how this work contributes to broader discourse in WASH research and intervention delivery and other cultural contexts. Below are three passages that exemplify how we have aimed to apply our learnings beyond rural India to sanitation interventions and programs in other parts of the world. We have added language to clarify that these insights reach beyond the local context. Added text has been underlined for easy identification:
[Lines 412-420] Though barriers may differ in other contexts, implementers of community interventions—in India and beyond—should ensure that recruitment extends to all; multiple and varied recruitment efforts are recommended. Further, activity venues and times should be accessible to all, or activities should be held in multiple venues and at different times to accommodate diverse groups. Separate activities for men and women also may improve reach of and engagement with intervention activities. Studies in India and Kenya have shown that women are less likely to speak up during community meetings if men are present [47,48]. In Sundara Grama, while the community meetings were intended to be sex-segregated, budget constraints allowed for only one co-ed meeting.
[Lines 448-456] Discourse around collective efficacy is growing in the WASH sector and new research provides insights into how researchers and practitioners can better understand the collective characteristics of the communities they engage, and subsequently design more effective community-based programming globally. Delea, et al (2018) have devised a scale to measure perceptions of collective efficacy, which includes factors like social disorder, social response, common values, social capital, social equity, community attachment, and agency, as potentially influential to the uptake of WASH behaviors [59,60]. Such a measure could help WASH program planners to better design, target, and evaluate their community-based interventions, although the measure may require adaptation to the local culture [57].
[Lines 457-464] Along with collective efficacy, specific focus might be needed on social cohesion within the WASH sector. Community-Led Total Sanitation (CLTS) is a well-established community-based sanitation intervention that relies on communities to collectively act in order to end open defecation. The CLTS Handbook states that the approach is more likely to succeed if implemented in communities that may already be socially cohesive (as indicated by small community size, homogenous population, a strong tradition of joint action, etc) [61]; a statement supported by a multi-country follow-up evaluation of different CLTS interventions that identified social cohesion as a factor in achieving sustained sanitation outcomes [62].
2) Lines 150-152: I was wondering if the collected and analyzed data were normally distributed. Could this be indicated in the article?
Response from authors: We did not analyze the data to assess if it was normally distributed as we did not deem that necessary or appropriate for the types of data collected. However, we have added mean age with standard deviation to Table 1 [Line 129].
|
|
Total N (%) |
Female N (%) |
Male N (%) |
|||
|
Total participants Intervention villages Non-intervention villages Average age in years |
152 65 87 45.8 |
(42.8%) (57.2%) SD=13.3 |
72 35 37 43.3 |
(47.4%) (53.8%) (42.5%) SD=11.0 |
80 30 50 48.0 |
(52.6%) (46.2%) (57.5%) SD=14.9 |
|
Caste1 |
152 |
|
|
|
|
|
|
General Caste Other Backward Castes Scheduled Caste Scheduled Tribe |
99 37 13 3 |
(65.1%) (24.3%) (8.6%) (2.0%) |
45 18 6 3 |
(45.5%) (48.6%) (46.2%) (100%) |
54 19 7 0 |
(54.5%) (51.4%) (53.8%) (0%) |
|
Latrine ownership2 |
152 |
|
|
|
|
|
|
Yes, functional Yes, non-functional No |
101 8 42 |
(66.4%) (6.0%) (27.6%) |
43 5 24 |
(42.6%) (62.5%) (57.1%) |
58 3 18 |
(57.4%) (37.5%) (42.9%) |
|
Frequency of latrine use for defecation (among latrine owners) |
110 |
|
|
|
|
|
|
Always Sometimes Never |
96 10 4 |
(87.2%) (9.0%) (3.6%) |
42 3 3 |
(43.8%) (30.0%) (75.0%) |
54 7 1 |
(56.3%) (70.0%) (25.0%) |
|
Attended the palla (intervention villages only) |
59 |
|
|
|
|
|
|
Yes No |
32 27 |
(54.2%) (45.8%) |
16 19 |
(50.0%) (70.4%) |
16 8 |
(50.0%) (29.6%) |
|
Attended the transect walk (intervention villages only) |
58 |
|
|
|
|
|
|
Yes No |
30 28 |
(51.7%) (48.3%) |
23 12 |
(76.7%) (42.9%) |
7 16 |
(23.3%) (57.1%) |
|
Attended the community meeting (intervention villages only) |
55 |
|
|
|
|
|
|
Yes No |
31 24 |
(56.4%) (43.6%) |
21 13 |
(67.7%) (54.2%) |
10 11 |
(32.3%) (45.8%) |
|
Attended the mother’s group meeting (intervention villages only) |
43 |
|
|
|
|
|
|
Yes No |
23 20 |
(53.5%) (43.6%) |
21 11 |
(91.3%) (55.0%) |
2 9 |
(8.70%) (45.0%) |
|
Attended the household visit (intervention villages and latrine owners only) |
23 |
|
|
|
|
|
|
Yes No |
20 3 |
(87.0%) (13.0%) |
6 3 |
(30.0%) (100%) |
14 0 |
(70.0%) (0%) |
If the reviewer is wondering specifically about the representativeness of our sample, we have discussed this in the discussion section, where we have identified some of the ways our sample differed from the sample in the full trial. Though we don’t have data on the population age distribution for the full trial sample, we suspect that the average age of our participants was higher than the trial average (45.8 years, SD=13.3), possibly because data collection took place during a time of day when more young people would be busy working. We also note that a much higher percentage of our participants were General caste than was estimated for the full trial population. Finally, latrine ownership was slightly higher in among our participants than the full trial population, possibly owing the higher percentage of General caste participants.
[Lines 502-510] Participants tended to be older (average age of all participants = 45.8 years, SD=13.3), with fewer participants from other age groups. This may have been influenced by the timing of the FGD, when younger adults might be busy working. The majority of participants (65%) were from General caste, which is likely not representative of village demographics, based on the larger trial data which found only 36% of households were General caste [40]. Household latrine ownership among participants (72.4% owners) was slightly higher than the larger trial data which found an average of 63.7% of households in intervention villages and 67.3% of households in control villages owned latrines [40]. This may be because of the higher percentage of General caste members in our sample.
3) On the average, how many interventions that participants have attended and in the same
vein the contrast between man vs woman. Would this have an impact on the overall
acceptance and perception of latrine use.
Response from authors: Participant attendance of Sundara Grama activities by sex is included in Table 1 [Line 129]. Because we were specifically recruiting villagers who had attended activities, we cannot say whether this was representative of the village. Process evaluation data from the full trial is currently being analysed and will report reach for each intervention activity for the trial villages specifically.
We did identify instances where women, especially young women, experienced barriers to attending the activities. We do not have data on how many other interventions participants had attended. In Results section 3.1.4, we explore participant perceptions of the role of “outsiders” (including interventions, government support and NGO projects) and their effect on latrine acceptance and use in the village. We did not identify a clear difference in responses between men and women, but this could certainly be a focus for further study.

Reviewer 2 Report
This is a very interesting manuscript that I enjoyed reading immensely and gained some new insights from. It describes the community’s perceptions of a sanitation behaviour change intervention. A paper describing the quantitative outcomes is separate, this paper focuses on the qualitative findings through focus groups with the communities, and as such it provides a rich addition to the literature. I particularly appreciated the comments on the “spillover” effect. RCT-type studies often worry about this so it is great to see it documented specifically here. My only comments are a few things which I found interesting which I would have liked to see further explored in the discussion. (1) What do the authors believe is the value of providing a behaviour change intervention without providing any physical intervention, as the communities requested? (2) How should interventions consider the increased spillover effects on women (line 342?)
Minor comments:
Lines 44-48 – this discusses open defecation and improved latrines without acknowledging the other rungs in the middle of the sanitation ladder.
Line 240 – what is the collector collecting? Tax, waste…?
Author Response
15 June 2020
To Whom It May Concern:
RE: ‘Community perceptions of a multi-level sanitation behavior change intervention in rural Odisha, India’(manuscript: ijerph-831499).
Many thanks to the reviewers who took the time to review our manuscript and give their feedback. Below is a point-by-point response to all comments.We have copied refered text sections into the responses below with line numbers, underlined all edits within text sections, and have also made all changes in track changes on the manuscript document for reviewing ease.Upon rereview, we have also made a few small edits to the manuscript.
We hope these revisions are met with your approval and appreciate your consideration of this manuscript. Please contact me if you have any questions.
Sincerely,
Bethany A. Caruso
REVIEWER 2
Comments and Suggestions for Authors
My only comments are a few things which I found interesting which I would have liked
to see further explored in the discussion.
(1) What do the authors believe is the value of providing a behaviour change intervention without providing any physical intervention, as the communities requested?
Response from the authors: First, we appreciate the reviewer’s time in providing comment to our manuscript.
Regarding the query: Though Sundara Grama was primarily a behavior change intervention, the overall intervention did include latrine repairs to select households in need, which unfortunately, were not completed by the time of this study to enable solicitation of feedback.
Sundara Grama was intentionally designed to be a behavior change intervention because high proportions of open defecators have been found to already own latrines and thus not need a physical intervetnion. As such, this research was responding to a specific need to encourage behavior change where a physical intervention was already available. That being said, in the process of designing the intervention, we did identify that some physical inputs would be critical to enablebehavior changes. For example, we learned that not all latrine owners had latrines that were functional. Therefore, we found it necessary to include minor latrine repairs into the interventions so that latrines people owned were actually usable. We also found limited access to water to be a barrier to latrine use. All toilets constructed as part of SBM were pour flush toielts requiring water. For many, water access is nearby and readily available, either from a pump, pond, or other source nearby, but others do not have access. Providing water systems to all would certainly facilitate latrine use, but is very costly and therefore outside the scope of what the intervention could afford to do. We recognize this as an overall limitation of the intervention. Additionally, in order to encourage safe disposal of child feces, we did provide mothers of children under age five with scoops and potties. Mother’s perceptions of this intervention, including he hardware provided, were explored separately and those findings are forthcoming.
Finally, community members definitely voiced the need for physical interventions, and we believe that this need should be considered in future intervention design. We have added the following text to the discussion:
[Lines 400-406] Some particpants found the intervention could not change sanitation behavior if improvements were also not made to sanitation facilities and water access.Indeed, behavior change interventions should take into account the physical needs of the community and determine if and how these needs can be met to enable behavior change. Latrine repairs were a part of the SundaraGrama intervention package because not all latrine owners had latrines that were functional or private to enable use. These had not been completed at the time of this study to determine if repairs made were deemed suitable.
(2) How should interventions consider the increased spillover effects on women (line 342?)
Response from authors: Thank you for raising this point. This is an interesting question and one that should be further explored in future research, possibly as a way to increase reach among women who may not be able to attend intervention activities. We have added the following paragraph to the discussion:
[Lines 477-489] Building social cohesion and social capital may also increase intervention spillover within the village, for those who could not attend activities, and to nearby villages. The women’s FGD in non-intervention village A was able to provide much more information about the intervention than the men’s FGD in the same village. This could be because women are less likely to spend their days working outside their village and therefore may be more plugged in to the village social network. One study in rural India has suggested that social networks may be a more effective target for public health interventions than village units [65]. To leverage social networks, interventions could identify and target established groups that have broad reach within the community as a means of diffusing messages to those who may be hard to reach. Such a tactic may be especially useful for reaching women who identified barriers to direct participation in intervention activities. Though community-level interventions may still be needed to achieve transformative WASH outcomes, more research into optimizing diffusion of interventionsthrough social networks may help to fill in the gaps in in intervention coverage and reach.
Minor comments:
Lines 44-48 – this discusses open defecation and improved latrines without acknowledging the other rungs in the middle of the sanitation ladder.
Response from authors: We have added information regarding the middle rungs in the sanitation ladder in India, Pakistan and Bangladesh, by adding the underlined text to the following paragraph:
[Lines 41-53] Despite benefits of sanitation, sustained use of latrines remains a challenge, particularly in India [15]. The 2019 WHO/UNICEF Joint Monitoring Programme (JMP) for Water Supply, Sanitation and Hygiene report estimated that 26% of India’s population practiced open defecation, 2% used an unimproved facility, 13% used limited (shared) sanitation facility, and 60% used at least a basic sanitation facility. Open defecation is most frequenly practiced in rural areas (36%) [16], and as per India’s National Family Health Survey (2015-16), collected just prior to the initiation of the present study, only 37% of rural households reported using an improved sanitation facility [17]. Comparatively, data from the WHO/UNICEF JMP report and World Bank showed that neighboring countries Bangladesh and Pakistan, despite having lower GDPs [18], had lower open defecation rates (<1% and 10%, respectively) [16]. Pakistan and Bangladesh had equal or lower percentages of their population using at least basic sanitation services compared to India (60% and 48% respectively), but had higher percentages of their populations using either an unimproved or limited sanitation facility [16].
Line 240 – what is the collector collecting? Tax, waste…?
Response from authors: The District Collector is the administrative head of the district. This individual monitors government programs (such as SBM) within the district and has the authority to impose fines/penalties or take actions against organizations and individuals for not adhering to the rules and regulations. District collectors do not necessarily “collect” anything, but because they have the ability to impose fines, they can be seen as creating a fear of punishment among villagers. We have added the following explanation (underlined) at the end of the quote:
[Line 245-249] One woman noted: “People should be pressurized that the district collector would come, then people will develop some fear and try to change,” (intervention village). The district collector is the administrative head of the district and has the authority to impose fines or penalties against organizations and individuals for not adhering to the rules and regulations, such as the regulations in SBM.
